# Factors Associated with the Acceptance of New Technologies for Ageing in Place by People over 64 Years of Age

**DOI:** 10.3390/ijerph19052947

**Published:** 2022-03-03

**Authors:** Sara Chimento-Díaz, Pablo Sánchez-García, Cristina Franco-Antonio, Esperanza Santano-Mogena, Isabel Espino-Tato, Sergio Cordovilla-Guardia

**Affiliations:** 1Department of Computer and Telematic Systems Engineering, Polytechnic School of Cáceres, University of Extremadura, 10003 Cáceres, Spain; schimento@unex.es (S.C.-D.); isabelet@unex.es (I.E.-T.); 2Health and Care Research Group (GISyC), University of Extremadura, 10003 Cáceres, Spain; pablosg@unex.es (P.S.-G.); esantano@unex.es (E.S.-M.); cordovilla@unex.es (S.C.-G.); 3Department of Medical-Surgical Therapy, Nursing and Occupational Therapy College, University of Extremadura, 10003 Cáceres, Spain; 4Nursing Department, Nursing and Occupational Therapy College, University of Extremadura, 10003 Cáceres, Spain

**Keywords:** ageing in place, aging in place, aging, technological development

## Abstract

Background: In the context of growing population ageing, technologies aimed at helping people age in place play a fundamental role. Acceptance of the implementation of technological solutions can be defined as the intention to use a technology or the effective use of it. Approaches based on the technology acceptance model (TAM) have been shown to have good predictive power for pre-implementation attitudes towards new technologies. Objective: To analyze the degree of acceptability of the use of new technologies for ageing in place and the factors associated with greater acceptance in people older than 64 years. Methodology: A descriptive cross-sectional study was carried out. Sociodemographic, clinical and environmental variables, architectural barriers, social risk and quality of life, degree of autonomy, morbidity, and risk of falls were collected in a population sample over 64 years of age in a large region of western Spain. The degree of acceptance of the use of technologies was measured through a scale based on the TAM. Results: Of the 293 people included in the study, 36.2% exhibited a high acceptability of new technologies, 28.3% exhibited a medium acceptability, and 35.5% exhibited a low acceptability. Of all the factors, age, education level, and living alone were significantly associated with high acceptance in the adjusted analyses. Conclusions: Younger age, a higher education level, and living alone are factors associated with a greater degree of acceptance of the use of technologies for ageing in place.

## 1. Introduction

It is estimated that 9% of the more than eight billion inhabitants on the planet are over 64 years old [1]. The prevalence of some type of disability in this age group is approximately 53% [1,2]. This percentage has increased exponentially in recent years, so much so that, according to the United Nations world population outlook report [3], in 2050, 1 in eleven people will be over 65 years of age, increasing this proportion to 25% in European countries. Spain currently ranks fourth among countries regarding population age, behind Japan, Italy and Germany, but in 2050, it could rank second, surpassed only by Japan, with a forecast of 22% of people over 60 years of age [4,5]. According to the projections by the National Institute of Statistics of Spain, in 2068, 29.4% of the rural population will be over 64 years of age, of whom more than 50,000 will live alone [2,6].

The substantial increase in the population group of older adults poses a challenge for governments and institutions due to the significant amount of resources required [7]. For this reason, efficient measures must be adopted to address the effects of population ageing and future needs, with institutional, social, economic, political, and cultural solutions that respond to the complexity of the phenomenon [8]. Living a long life is considered a relevant characteristic of only contemporary developed societies [9]. As people age, in general, the need for health care increases because there is a relationship between ageing and the worsening or resurgence of pathologies [10]. The challenges of an increasingly ageing society make it necessary to develop a social model of sustainable care that promotes autonomy and self-care for as long as possible [11]. This is the philosophy pursued by the ageing in place movement [12]. The WHO defines ageing in place as satisfying people’s desire and ability, by providing appropriate services and assistance, to remain living relatively independently in the community, either in their current home or in appropriate housing, avoiding or delaying a traumatic relocation to a centre for dependents, such as a nursing home [13]. Along with the level of dependence, the degree of institutionalization in many cases is conditioned by financial problems and fundamentally supported by the uncertainty of the remaining life of the person [14].

Ageing in place is a very broad and complex conceptual matrix that encompasses both the physical and social environments, with an inherent psychological component, containing a unique importance and meaning for each person [15]. The “home” can be defined as a space with people, practices, objects, and representations that can have several different characteristics, which include geographic location and belongings, but that is invested with a meaning and value deeply connected to the person [16]. The key to successfully ageing at home can be associated with social capital, which includes both social networks and social and economic resources, in addition to the accessibility of the built environment [15]. In this sense, it has been suggested that care expectations may be conditioned by habitual intergenerational contact [17]. This conflicts with an increasingly generalized context of loneliness [18]. Previous studies indicate that given that the majority of older adults prefer to remain in their homes as they age instead of moving to an institution or going to live with family members [19], the paradigm that comprises ageing in place could be essential to enabling a greater life expectancy for elderly individuals [15]. However, this option raises several challenges and doubts about the impact on various components related to the health of elderly individuals [20]. Being able to choose a place of residence seems to be related to a higher quality of life [21]. However, as autonomy is lost, there are no conclusive studies to support any particular model of care for functionally dependent older people [22]. In this situation, the option of home care seems to be associated with a higher quality of life and perceived functionality, and there is evidence that shows a possible increase in morbidity, mortality, and hospital admissions when compared to institutionalization [22], as well as an association with a greater burden on caregivers [23]. In addition, the health crisis caused by SARS-CoV-2 has contributed to people becoming more vulnerable to community residential contexts [24].

The factors that seem related to meeting the expectations of ageing in place are encompassed in various areas, among which are the social and economic situation, functionality (physical, cognitive, etc.), mental health, the environment, and perceived quality of life [25]. Although it seems that older age is associated with lower expectations of achieving a successful ageing model, and that having low expectations, in turn, is related to not believing that it is important to seek health care [25], the association of expectations with the factors that are related to the successful implementation of ageing in place has not yet been sufficiently explored in our context. Thus, in addition to the aforementioned factors, such as level of dependence and health conditions (comorbidities and medication consumption), the social environment, economic and social resources, architectural barriers in the place of residence [22], other factors such as fear of falling [26], and those related to lifestyle habits, such as alcohol consumption [26] or physical activity [27], could condition care expectations in an ageing in place model. Exploring these associations could be essential to guide efforts, optimizing resources with targeted interventions. In this sense, technological solutions are becoming increasingly important as a factor that can be fundamental to facilitate ageing in place [28].

The characteristics of the solutions applicable to ageing in place, such as innovations in technologies for the diagnosis, prevention, monitoring, and treatment of chronic conditions, can improve care, increase quality of life, and substantially reduce the emotional and financial costs associated with ageing [28]. These devices include solutions as varied as vision and hearing aid devices, furniture or appliances for daily living, aids to improve mobility, and environmental adaptations [29]. In addition to existing solutions such as telecare services that help beneficiaries receive care through the use of communication and information technologies, with immediate responses to emergencies or to insecurity, loneliness and isolation [30], new technologies for the home can improve the safety and care of older adults through the monitoring of their environments with sensors that detect changes such as falls, or the opening of the medicine cabinet door or front door, providing proactive services in the home [31]. These sensors could help to recognize and monitor basic activities of daily living (walking, bathing, and dressing) and instrumental activities (such as cooking, driving, and using the telephone and the computer), recognizing patterns that reflect the physical and cognitive health conditions of older adults and any deviation that may indicate problems which require intervention [32].

In addition to sensors in the home, there is evidence that monitoring devices directly connected to elderly individuals can improve clinical outcomes for pathologies such as diabetes and heart failure, among others [28]. Additionally, voice assistants are an artificial intelligence technology that is becoming more affordable and can help ageing in place and potentially alleviate caregiver burden [33]. Furthermore, the use of social media technologies is gaining strength in the context of ageing in place [34], thanks to its potential to mitigate loneliness.

Although the use of new technologies can contribute to greater security in the home of older adults and to ageing in place, these new environments can generate doubt because little is known about how older people perceive assistive and monitoring technologies [29]. Aspects such as privacy, perceived need, lack of training, or cost are elements that have been associated with the failed implementation of so-called “gerontechnologies” [29].

The acceptance of the use of technological solutions can be defined as the intention to use or effective use of a technology [35]. Knowing the factors that influence this acceptance should be prioritized ahead of implementing ageing in place technologies [36]. The pre-implementation acceptance of technological solutions is an effort to understand the use of modern technology by older adults prior to its incorporation. For this purpose, researchers more frequently resort to approaches that involve the technology acceptance model (TAM) [35] based on the theory of reasoned action [37], with the objective of predicting the behaviour of people based on their attitudes and intentions.

There is strong evidence that the TAM predicts the subsequent use of a new technology [38,39]. The key areas that are assessed in the TAM are perceived usefulness (PU) [40], defined as the feeling that users have about how the task they perform with the device when compared to not using it; perceived ease of use (PEU) [35], which refers to the degree to which an individual believes that using a particular system is effortless; attitude towards use (ATU) [39], defined as the positive or negative feelings held by users of a technology; intention towards use, which is defined as the degree to which a person has made conscious plans to develop (or not) some future behaviour. This intention has been estimated by some authors using Relevance (Re) as a cognitive proxy of the intention of use [41]. Some studies have been able to predict more than 70% of the intention to use a new technology by adding additional factors to the previous elements, such as social influence and identifying four factors as moderators (sex, age, experience, and ATU) [36].

This theory on the acceptance and use of new technologies has strongly penetrated the literature on the use of new technologies for health because the TAM is, in the industry outside of health care, somewhat a gold standard [42]. The literature comprises a multitude of studies focusing on users of the TAM [42,43,44,45], which is becoming the most commonly used method for studies on the adoption of new technologies [43]. However, there are still important gaps in knowledge regarding the role of other factors related to social and personal contexts and how these positively influence greater acceptance [42].

Knowing which factors influence how well older adults accept technologies under the premise of meeting care expectations in an ageing in place environment and how these factors are related to health and environmental conditions and the products and services available to elderly individuals could shed light on how efforts should be directed in a context of an ever-ageing population. Therefore, the aim of this study was to analyze the degree of acceptability of the use of new technologies for ageing in place and the factors associated with a greater degree of acceptance by people older than 64 years of age.

## 2. Materials and Methods

### 2.1. Design and Participants

A cross-sectional study was carried out using a population older than 64 years of age residing in Extremadura, a large region in western Spain, during the months from December 2020 until July 2021. To recruit participants, the Foundation for the Training and Research of Health Professionals in Extremadura (Fundación para la Formación e Investigación de los Profesionales de la Salud en Extremadura, FundeSalud (Mérida, Spain)) and the Extremadura Service for the Promotion of Autonomy and Care for Dependence (Servicio Extremeño de Promoción de la Autonomía y Atención a la Dependencia, SEPAD (Mérida, Spain)), which participated in the project “4IE+ Instituto Internacional de Investigación de Innovación del Envejecimiento (4IE+ International Institute for Innovation Research on Ageing), allowed access to social resources for sample collection. As an exclusion criterion, participants could not present cognitive impairment, which was established if 4 or more fails were identified in the Pfeiffer Short Portable Mental Status Questionnaire (SPMSQ). It is an easy-to-administer test with high sensitivity and specificity (91 and 90%, respectively) [46].

### 2.2. Data Collection Instrument

To carry out the data collection, a computer-aided hetero-administered questionnaire, composed of scales validated in the Spanish population, was developed. The interviewers who carried out the sample collection received specific training on the use of the questionnaire through portable devices (tablets). In addition, a pilot sample of 80 interviews was carried out to detect difficulties in sample collection and to propose points for improvement. The questionnaire was designed following conditional logic to guarantee the quality of the data collected, preventing errors and inconsistencies.

### 2.3. Measures

#### 2.3.1. Main Study Variable

The degree of acceptability of the use of new technologies was measured as the main variable of the study through questions based on the TAM [47] and divided into 4 dimensions: PU, PEU, ATU, and Re (Appendix A). The TAM questions were specifically adapted for ageing in place assistive technologies. The possible scores range from 11 (lowest acceptance) to 55 (highest acceptance). To facilitate interpretation, in addition to being a quantitative variable [39,48], the scale was categorized into 3 degrees: low, medium, and high acceptability, by tertiles, based on the total score obtained and for each of the 4 dimensions (PU, ATU, PEU, and Re). The TAM has good validity and reliability, with Cronbach’s alpha values greater than 0.8 and [39].

#### 2.3.2. Independent Variables

Sociodemographic and personal variables were age, sex, body mass index (BMI; categorized as underweight, BMI < 18.5. kg/m^2^; normal weight, 18.5–24.9 kg/m^2^; overweight, 25–29.9 kg/m^2^; obese, BMI ≥ 30 kg/m^2^), place of residence, and marital status. To assess the social and family situation and to detect risk situations and/or social problems, the Gijón Scale was used. This widely validated scale consists of 5 items that evaluate family, finances, housing, relationships and social support; based on the overall score, social risk is classified as low (<10), medium (10 to 16), and high (≥17). The scale has an acceptable degree of validity and excellent reliability (intraclass correlation coefficient of 0.957) [49]. Comorbidities were assessed using the Self-Administered Comorbidity Questionnaire (SCQ). This validated scale has a reliability of 0.94 (interclass correlation coefficient) and Spearman correlation coefficient of 0.81, presenting advantages in terms of efficiency relative to the Charlson index (reference scale), with a Spearman correlation coefficient of r = 0.55 [50].

To assess the risk of falls, a fall risk index (Downton) and fear of falling instrument (Falls Efficacy Scale International, FES-I) were used. The Downton scale has a sensitivity of 0.58 and a specificity of 0.62 [51]. The FES-I evaluates physical activity and fall risks mainly at home. The scale FES-I has excellent internal reliability and test–retest reliability (Cronbach’s alpha = 0.96; ICC = 0.96) [52]. When the score on the scale is three or more points, the person is considered to be at high risk of falling.

The consumption of alcohol and other drugs was assessed with a 30 day approximation using Timeline Followback Method Assessment [53]. For alcohol consumption, risk consumption was determined using the Audit-C scale [54], a widely validated scale with a sensitivity between 54 and 98% and a specificity between 57 and 93% [55].

Data on the quantity and quality of physical activity performed during free time (leisure and home maintenance activities) were obtained through the shorter version of the Minnesota Leisure Time Physical Activity Questionnaire (VREM) [56]. The test–retest reliability of the scale is 92.5%, with a Kappa index of 0.88 (95% CI: 0.79 to 0.97) and an intraclass correlation coefficient of 0.96 (95% CI: 0.95–0.98). The cut-off points are: Sedentary < 1250 METS-min/14 days; Moderately active 1250–2999 METS-min/14 days; Active 3000–4999 METS-min/14 days; Very active ≥ 5000 METS-min/14 days [56].

The degree of independence for activities of daily living was assessed using the Barthel Index, an internationally validated scale capable of assessing the degree of dependence and the areas where applied. This scale, validated in Spanish, has good reliability (with Cronbach’s alpha greater than 0.70), and its structural validity discriminates between groups and detects changes over time [57]. In the event that a person presented some degree of dependence, the presence of the primary caregiver was assessed [58]. The cut-off points for the level of dependency score are: <20 Total; 20–35 Severe; 40–55 Moderate; ≥60 Mild; 100 Independent.

Expectations of care were explored in terms of place of residence, degree of institutionalization, and presence of care. Expectations of care included the preferences of the person when ageing at home, in residential centres, at day centres, or having the help of a primary caregiver, either a family member or not [10,59].

Health-related quality of life was assessed using the Euroqol scale (EQ-5D-5L) [60]. This generic and standardized instrument consists of 2 parts: (i) the EQ-5D descriptive system and (ii) a visual analogue scale (VAS). The EQ-5D descriptive system includes 5 dimensions: mobility, self-care, usual activities, pain/discomfort, and anxiety/depression. In addition, EQ-5D health states were directly converted into single index values specific to our country (Spain) using a specific set of values [61]. For the VAS, individuals scored their health between 2 extremes, 0 and 100, the worst and best imaginable health status, respectively. The EQ-5D-5L has shown excellent validity and reliability as a health measure, improving the ceiling effect of its 3-level predecessor [62].

Data were collected on accessibility in the home through the presence of architectural barriers (stairs and/or steps) at the access door of the building or home and inside the building or home. In homes, the presence of stairs and/or steps in different rooms, the presence of a bathroom with a shower tray, and other barriers and/or adaptations were assessed. In addition, questions were asked about access to resources (telecare, home help, 24 h home help, physiotherapy, speech therapy, occupational therapy, psychosocial habilitation–cognitive stimulation or others), as well as the use of technical aids or support products [63].

### 2.4. Data Analysis

Descriptive analyses were performed to study the distribution of the variables. The normality of the distribution of the quantitative variables was verified using measures of central tendency and dispersion; the mean and standard deviation (±SD) are reported for data that had a normal distribution, and the median and interquartile range (IQR) are reported for data that did not. The Pearson chi-square test was used to compare categorical variables. Quantitative sociodemographic variables were compared between groups by the Student’s *t*-test for variables with a normal distribution and with the Kruskal–Wallis test for variables with a non-normal distribution. To quantify the strength of association between the measured variables and the degree of acceptance, first, crude odds ratios (cORs) were obtained between each variable and the degree of acceptance (low acceptance as a reference category). In a second step, to adjust the effect between the variables, multivariate analysis was carried out using multinomial logistic regression with the degree of acceptance categorized into the 3 levels (low acceptance as a reference category) as the dependent variable and the rest of the factors as independent variables in the model, which allowed obtaining adjusted odds ratios (aORs) with their corresponding 95% confidence intervals (95% CIs). The analyses were performed using SPSS 25.0 for Windows (SPSS, Chicago, IL, USA). We considered significant *p*-values < 0.05.

#### Ethical Aspects

All participants included in the study agreed to participate by signing an informed consent form. The ethical principles outlined in the Declaration of Helsinki were respected at all times [64]. The study protocol was approved by the Research Ethics Committee of the University of Extremadura (Cod. 89/2020).

## 3. Results

A total of 339 people were invited to participate in the study, of whom 321 agreed to participate. However, 28 people were excluded due to cognitive impairment. The median age in years (68 years, IQR (65–72)) of the people who refused to participate was 9 years younger than the sample of participants (*p* < 0.001), and 52.9% were women in the excluded sample. A difference was found in terms of gender distribution in the sample finally analyzed (*p* = 0.433).

In the characteristics of the entire sample (n = 293) finally analyzed (Table 1), we found that the median age was 76 (70–86) years, and they were mostly women (61.1%) with primary education (61.1%). Most of the sample was overweight or obese (71%). The risk of falls in the entire sample was mostly low (63.1%) and almost no hospitalizations or falls were reported in the last year.

Regarding alcohol consumption (Table 1), consumption was minimal, with a median of 0.00 (0.00–1.00), distributed as non-risk consumption for 282 (96.2%) participants and risk consumption for 11 (3.8%) participants, with scores equal to or greater than four. No cases of consumption of other drugs or non-prescription medications were reported. 38 respondents (13%) had some degree of dependence. Only 38 of the respondents (13%) had some degree of dependence and 87 had a medium and high social risk. On the other hand, the median quality of life scores obtained for each dimension (mobility, self-care, daily activities, pain and discomfort, and anxiety and depression) was 1.00 (1.00–2.00), with a total score of 70.00 (50.00–90.00) and an EQ-5D-5L index score of 0.91 (0.62–1.00).

Of the total sample that lived in their home (n = 201; 68.6%), 47 (16%) had barriers to accessing their building, 46 (15.7%) had barriers to accessing their home, 52 (17.7%) had barriers inside their home, and 79 (27%) had stairs in different rooms of the home. Of these, 13 (4.4%) had accessibility adaptations in all cases, and for 24 (8.2%), although they had adaptations, they were not complete. Last, 154 (52.6%) had a shower tray instead of a bathtub.

The median TAM score for the entire sample was 23.00 (14.50–46.00) (PU: 4.00 (2.00–10.00); PEU: 9.00 (6.00–15.00); ATU: 7.00 (3.00–11.00); Re: 6.00 (3.00–15.00)). When categorizing by tertile, 106 (36.2%) participants had high acceptability of new technologies, 83 (28.3%) had medium acceptability, and 104 (35.5%) had low acceptability. The levels of acceptability for the different TAM dimensions were distributed as follows: PU (38.6% high, 14.7% medium, and 46.8% low), PEU (42.3% high, 33.4% medium, and 24.2% low), ATU (29.4% high, 28.0% medium, and 42.7% low), and Re (35.5% high, 15.4% medium, and 49.1% low).

When comparing the median age based on level of acceptance of new technologies in the four dimensions of the TAM scale (Figure 1), for all dimensions, the median age was significantly lower as the degree of acceptance increased.

When dividing the sample based on the degree of acceptability of the use of new technologies (TAM score) and comparing the sociodemographic and personal characteristics (Table 2), married individuals (58.5%) in the group had a greater readiness to use new technologies, and widowed individuals were predominant in the group with least acceptance (54.8%). Additionally, there were distribution differences in the level of education, where the percentage of participants without education was ostensibly lower in the high-acceptance group than in the middle and low-acceptance groups (7.5 vs. 20.5% and 36.5%, respectively).

When the physical variables were compared based on the degree of acceptance (Table 3), there was a lower risk of falls (81.1% low risk) in the high-acceptance group than in the low-acceptance (54.8%) and medium-acceptance (50.6%) groups (*p* < 0.001). We also found differences in hospitalizations in the last year; 10.4, 23.1, and 15.7% of participants in the high-acceptance group, low-acceptance group, and medium-acceptance group, respectively, were hospitalized for at least 1 day in the last year (*p* = 0.015). When comparing the number of comorbidities, the median (IQR) for the high-acceptance group (2.00 (1.00–2.00)) was different from that (2.00 (1.00–3.00)) for the other acceptance groups (*p* = 0.005). In addition, risky consumption of alcohol was more prevalent in the group with a higher degree of acceptance of technological solutions (7.5%) (medium-acceptance group, 3.6%; low-acceptance group, 0.0%) (*p* = 0.009).

When the presence of architectural barriers among respondents who resided at home was analyzed based on the degree of acceptance of new technologies (Table 4), there was a lower percentage of some barrier among those in the high acceptance (46.7%) than among those in the medium-acceptance group (60.4%) and the low-acceptance group (68.6%) (*p* = 0.032). However, we did not find differences among the groups when the different types of barriers outside and inside the home were compared (Table 4). There were also no differences in the distribution of users who had home adaptations (n = 79), with six (25%) in the low-acceptance group, seven (35%) in the medium-acceptance group, and eleven (31.4%) in the high-acceptance group (*p* = 0.856).

When comparing the degree of dependence based on acceptance (Table 5), there was a greater degree of independence regarding activities of daily living in the high-acceptance group, with 103 participants (97.2%) totally independent, compared to 64 (77.1%) in the medium-acceptance group, and 88 (84.6%) in the low-acceptance group (*p* = 0.002). Regarding quality of life (EQ-5D-5L) based on the degree of acceptance of new technologies, there were significant differences in all dimensions of the scale (Table 5), with a clearly higher quality of life index in the high-acceptance group (1.00 (0.91–1.00)) than in the low-acceptance (0.78 (0.47–1.00)) and medium-acceptance (0.78 (0.52–1.00)) groups (*p* < 0.001). We also found differences in the VAS score, with an increase of 10 points for every level of increased acceptance (*p* = 0.003).

When analyzing the crude and adjusted strengths of association between the variables assessed and the degree of acceptance (Table 6), age was significantly associated with a medium and high acceptability to new technologies, with greater acceptability associated with a younger age, a finding that was maintained for both levels of acceptability in the adjusted analyses, with a mean acceptability aOR of 0.94 (95% CI: 0.90–0.97; *p* < 0.001) and a high acceptability aOR of 0.88 (95% CI: 0.84–0.92; *p* < 0.001). A higher education level was associated with a medium level of acceptability, with an aOR of 2.82 (95% CI: 1.09–4.80; *p* = 0.029) for a secondary education. The highest acceptability group showed strong magnitudes of association for both primary and secondary education, with aORs of 7.23 (95% CI: 2.14–27.88; *p* = 0.002) and 5.30 (95% CI: 1.90–14.78; *p* = 0.001), respectively. Care expectations were not associated with any level of acceptability in the crude or adjusted models. For the other variables, despite finding a significant strength of association in the crude models, only the category of living was associated with a high degree of acceptability when adjusted for the other variables (aOR: 5.11 (95% CI: 1.33–19.55); *p* = 0.017).

## 4. Discussion

The results of this study indicate that a younger age, higher education level, and living alone are the factors that are associated with a greater readiness to use new technologies for ageing in place. Although all the factors measured, with the exception of sex and care expectations, show significant crude magnitudes of association with high acceptability, the adjusted models indicate that these associations are confounded by age, education level, and living alone.

Adapting technological solutions to the characteristics of the intended end users is decisive for technological solutions to be accepted and incorporated into the routine and context of the users [42]. The findings of our study suggest that when proposing technological solutions for ageing in place, the factors that must be considered prior to implementing them are those related to sociodemographic characteristics, such as age, education level, or living alone. Younger age and higher education level are predisposing factors to the adoption of different technologies for ageing in place, which is also frequently found in the literature [36]. Thus, the same results were observed in terms of age and education level when implementing activity tracking systems [65] or when the acceptance of electronic security devices for ageing in place was analyzed, adding living in a rural area as a positive factor for successful implementation [66].

Other quantitative studies that analyzed different technological solutions for ageing in place report that other variables, such as subjective health status, have a significant influence on the pre-implementation phase [36]. In our study, we analyzed perceived quality of life, comorbidities, degree of autonomy, risk of falls, degree of institutionalization, and architectural barriers in the home, and although in the crude models all these factors showed significant magnitudes of association with a high degree of acceptance, the adjusted models indicated that these relationships can only be explained by the association of these factors with age, which seems to be, in turn, strongly associated with acceptance. According to our results, each year of increased age in potential users of new ageing in place technologies decreases the high acceptance of use by 12%, appearing as a negative factor for acceptance in all dimensions of the TAM scale. The absence of multivariate analyses applied in our study could explain the associations between subjective health status and acceptance reported by other studies [65,66,67].

The dimensions used to allow the dimensions to be assessed individually for each individual, however, some models encompass the dimensions and allow the dimensions to be assessed in a grouped way for the population, such as the Unified Theory of Acceptance and Use of Technology (UTAUT). UTAUT and TAM both are models that serve for individual acceptance and organizational acceptance over the IT components by the theory of reasoned action, the technology acceptance model, the motivational model, the theory of planned behaviour, a combined theory of planned behaviour/technology acceptance model, the model of personal computer use, diffusion of innovations theory, and social cognitive theory. TAM is mostly individual level [68].

We found a significant association between physical activity and moderate acceptance. Physical activity has been reported in the literature as a favourable factor for the use of new technologies [69]. Physical activity scales assess activities such as walking 30 min every day or performing tasks such as “gardening” or activities that involve participation in the community [70]. In addition, vigorous activity is related to greater community involvement because it leads to a better state of health. Based on some research, people with a higher level of physical activity have more technological devices that support such activities [71]. Physical activity could be understood as a facilitating factor for the acceptance of ageing in place technologies because the accessibility to some technological devices encourage individuals to maintain their physical capacities.

Our results indicate that 64.5% of our study population has a medium–high readiness for the use of new technologies aimed at facilitating ageing in place. This seems to be a high percentage, considering that the average age of our sample was 76 years. Younger age is considered a facilitator for the use of new technologies [72] due to its influence on the perceptive, motor, and cognitive abilities of individuals [73], in addition to the importance of compensation processes that older people develop to adapt to changes, and the crucial role that motivation, attitude and experience play in all social interactions [73]. Taking into account the population ageing trend [3], the challenge of promoting the use of *ageing in place* technologies that are accepted and used involves a deep understanding of the role played by the different factors involved. In this sense, the finding that a higher education level is associated with a greater readiness to use these technologies, despite the negative effect of age, is a promising aspect in our context because as new generations reach an age of 65 years or older, they would have a higher level of education and knowledge about the use of technologies [74]. This finding is consistent with that reported in another studies [28] that related the level of education with a greater success in adopting gerontotechnologies aimed at facilitating ageing in place. On the other hand, increasing the acceptability of new technologies must be a priority challenge in increasingly ageing societies [28] if the aim is to opt for an ageing in place model. In this sense, it is necessary to promote interventions aimed at increasing the readiness and predisposition to use these technologies in the elderly population. Apart from the study of the factors associated with the acceptance of new technologies as addressed in this study and other works [29], a better understanding of the reasons why an older person might use ageing in place enabling technologies could be a good starting point to guide these initiatives [75].

In general, older people prefer to continue living in their own homes instead of moving to residential care institutions [29]. Therefore, it is not surprising that the results of our study show that, regardless of age and education level, people who live in their home show a greater acceptance of using new technologies, especially those who live alone. Assistive technologies and sensors in the home environment and/or body systems that track the movement of individuals can contribute to a greater sense of security in the home [29]. The positive effects of ageing in place technological devices on improving social networks, independence, psychological well-being, and social status of older adults can allow these individuals to stay at home when living with challenges related to age, such as falls, isolation, medication management, sensory impairment, decreased mobility, and the possibility of negative consequences [76]. Living alone is not synonymous with loneliness [77]; however, it is a predisposing factor, especially in very old, rural, and sparsely populated regions [78]. A significant number of studies have explored the positive effects of computer-mediated communication on creating social networks for and improving the independence, psychological well-being, and social status of older adults [76]. However, there are also drawbacks to online social support, for example, the absence of physical contact, the lack of auditory and visual contextual signals, the desire for more social contact, and not being able to offer necessary help to other people [79].

Owing to the number of factors measured in our study and the analyses carried out, the results of this work shed light on how, a priori, all the factors we have identified in the literature could be related to accepting new technologies. However, due to the cross-sectional nature of the study, the results should be interpreted with caution. Notably, as shown by some qualitative studies [36], the cultural component of these associations may be important; therefore, it is not possible to guarantee the external validity of our results.

## 5. Conclusions

Younger age, a higher education level, and living alone are the factors that are most associated with greater acceptance of using ageing in place technologies. In all dimensions of the technology acceptance scale (TAM), older age appears to be related to lower scores. The crude associations found with other factors, such as perceived quality of life, comorbidities, degree of autonomy, and risk of falls, ceased to be associated in the adjusted model because of the confounding factors of age, education level, and place of residence. Physical activity was associated with a moderate degree of acceptance but not with a high degree of acceptance. Other variables, such as sex and expectations of care, did not show associations with acceptance, in neither the crude nor adjusted models.

The results of this study can help guide actions aimed at promoting the use of technologies that facilitate ageing in place in the context of an ageing population. In-depth knowledge of the pre-implementation facilitating factors of these technologies could help to optimize these initiatives. However, more research is needed that takes into account cultural contexts. Studies that confirm that acceptance leads to the actual implementation in this context are also necessary.

## Figures and Tables

**Figure 1 ijerph-19-02947-f001:**
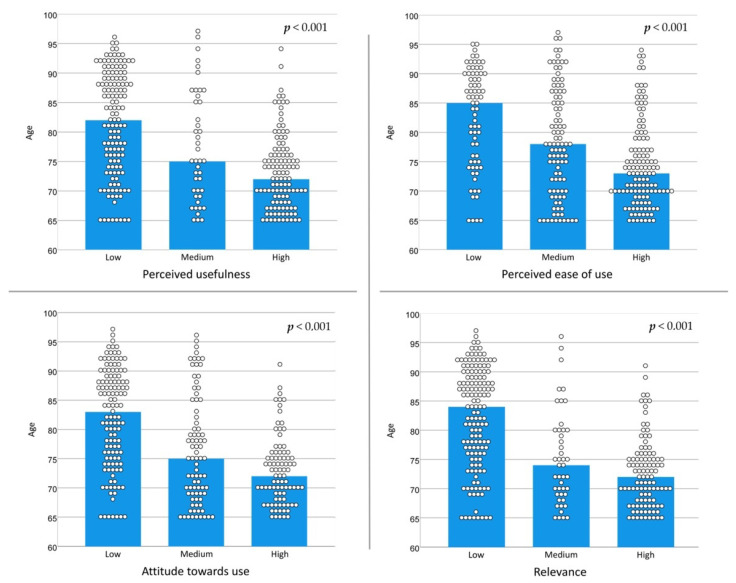
Age (median) and the degree of acceptance of new technologies in the dimensions of the TAM. TAM score: low, 0 to 18.3; medium, 18.4 to 36.6; high, 36.7 to 55.

**Table 1 ijerph-19-02947-t001:** Characteristics of the entire sample (n = 293).

Sociodemographic and Personal Variables	
Age Median (IQR)	76 (65–97)
Sex n (%)	
Men	114 (38.9)
Women	179 (61.1)
Marital Status n (%)	
Single	29 (9.9)
Married	122 (41.6)
Widowed	123 (42)
Divorced	19 (6.5)
Education n (%)	
Uneducated	64 (21.8)
Primary education (completed or unfinished)	179 (61.1)
Secondary education (completed or unfinished)	32 (10.9)
University (completed or unfinished)	18 (6.1)
Physical variables	
BMI Median (IQR)	25.54 (24.54–31.11)
BMI n (%)	
underweight	1 (0.3)
normal weight	84 (28.7)
overweight	116 (39.6)
obese	92 (31.4)
Days of hospitalization Median (IQR)	0 (0–0)
Falls Median (IQR)	0 (0–0)
Alcohol consumption Median (IQR)	0.00 (0.00–1.00)
AUDIT-C n (%)	
Non-risk	282 (96.2)
Risk	11 (3.8)
Dependence, social risk and institutionalization	
Barthel Median (IQR)	100 (90–100)
Degree of dependence n (%)	
total dependence	9 (3.1)
severe dependence	11 (3.8)
moderate dependence	14 (4.8)
mild dependence	4 (1.4)
independent	255 (87)
Social risk Median (IQR)	9.00 (8.00–11.00)
Social risk n (%)	
normal	109 (37.2)
medium	84 (28.7)
high	3 (1.0)
Level of institutionalization	
Lived at home with someone (not go to a day center)	93 (31.7)
Lived at home with someone (go to a day center)	48 (16.4)
Lived alone (not go to a day center)	44 (15)
Lived alone (go to a day center)	11 (3.8)
Lived in a nursing home	97 (33.1)

IQR: Interquartile Range; BMI (Body Mass Index); AUDIT-C: Risk alcohol consumption.

**Table 2 ijerph-19-02947-t002:** Sociodemographic and personal variables according to the level of acceptability of the use of new technologies.

	Low (n = 104)	Middle (n = 83)	High (n = 106)	*p* Value
Sex n (%)				
Male	39 (37.5)	36 (43.4)	39 (36.8)	0.614 ^1^
Female	65 (62.5)	47 (56.6)	67 (63.2)	
Marital Status n (%)				
Single	10 (9.6)	12 (14.5)	7 (6.6)	**<0.001 ^1^**
Married	31 (29.8)	29 (34.9)	62 (58.5)	
Widowed	57 (54.8)	37 (44.6)	29 (27.4)	
Divorced	6 (5.8)	5 (6.0)	8 (7.5)	
Education n (%)				
Uneducated	38 (36.5)	17 (20.5)	8 (7.5)	**<0.001 ^2^**
Primary	55 (52.9)	52 (62.7)	72 (67.9)	
Secondary	10 (9.6)	7 (8.4)	15 (14.2)	
University	1 (1.0)	6 (7.2)	11 (10.4)	
DK/DA	-	1 (1.2)	-	

^1^ Pearson’s chi square; ^2^ Fisher’s exact test; IQR: Interquartile Range; Significant *p* values (<0.05) are in bold.

**Table 3 ijerph-19-02947-t003:** Physical variables based on the level of acceptability of the use of new technologies.

	Low (n = 104)	Middle (n = 83)	High (n = 106)	*p* Value
BMI n (%)				
Underweight	1 (1)	-	-	0.143 ^1^
Normal Weight	35 (33.7)	17 (20.5)	32 (30.2)	
Overweight	34 (32.7)	42 (50.6)	40 (37.7)	
Obese	34 (32.7)	24 (28.9)	34 (32.1)	
Risk of Falling n (%)				
Low	57 (54.8)	42 (50.6)	86 (81.1)	**<0.001 ^1^**
Medium	4 (41.3)	3 (3.6)	3 (2.8)	
High	43 (41.3)	38 (45.8)	17 (16.0)	
Hospitalized in the last year n (%)	24 (23.1)	13 (15.7)	11 (10.4)	**0.015 ^2^**
Falls in the last year n (%)	27 (26)	14 (16,9)	16 (15,1)	0.108 ^2^
Comorbility Median (IQR)	2.00 (1.00–3.00)	2.00 (1.00–3.00)	2.00 (1.00–2.00)	**0.005 ^3^**
AUDIT-C n (%)	-	3 (3.6)	8 (7.5)	**0.009 ^2^**

^1^ Fisher’s exact test; ^2^ Pearson’s chi square; ^3^ Kruskal Wallis test; BMI (Body Mass Index); AUDIT-C: Risk alcohol consumption IQR: Interquartile Range; Significant *p* values (<0.05) are in bold.

**Table 4 ijerph-19-02947-t004:** Architectural barriers in the home.

	Low (n = 52)	Middle (n = 49)	High (n = 100)	*p* Value
Some type of architectural barrier n (%)	35 (68.6)	29 (60.4)	43 (46.7)	**0.032** ** ^1^ **
Type of architectural barrier n (%)				
Access door to the building	17 (32.7)	12 (24.5)	18 (18.0)	0.125 ^1^
Inside the building up to the front door of the dwelling	15 (28.8)	11 (22.4)	20 (20.0)	0.467 ^1^
Home	16 (30.8)	16 (32.7)	20 (20.0)	0.163 ^1^
Stairs/step	24 (46.2)	20 (40.8)	35 (35.0)	0.397 ^1^
Shower tray	37 (71.2)	36 (73.5)	81 (81.0)	0.331 ^1^
Other Barriers	2 (3.8)	2 (4.1)	-	0.062 ^2^

^1^ Pearson’s chi square; ^2^ Fisher’s exact test; IQR: Interquartile Range; Significant *p* values (<0.05) are in bold.

**Table 5 ijerph-19-02947-t005:** Dependence, social risk, and quality of life based on the degree of acceptance of new technologies.

	Low (n = 104)	Middle (n = 83)	High (n = 106)	*p* Value
Degree of dependence n (%)				
Total	3 (2.9)	6 (7.2)	-	**0.002 ^1^**
Severe	4 (3.8)	6 (7.2)	1 (0.9)	
Moderate	7 (6.7)	6 (7.2)	1 (0.9)	
Mild	2 (1.9)	1 (1.2)	1 (0.9)	
Independent	88 (84.6)	64 (77.1)	103 (97.2)	
Social Risk n (%)				
Low	25 (50.0)	26 (55.3)	58 (58.6)	0.265 ^1^
Medium	24 (48.0)	19 (40.4)	41 (41.4)	
High	1 (2)	2 (4.3)	-	
EQ-5D-5L Medium (IQR)				
Dimensions				
Mobility	2.00 (1.00–3.00)	2.00 (1.00–3.00)	1.00 (1.00–1.00)	**<0.001 ^2^**
Self-Care	1.00 (1.00–3.00)	1.00 (1.00–3.00)	1.00 (1.00–1.00)	**<0.001 ^2^**
Act. Daily Living	1.00 (2.00–3.00)	1.00 (2.00–3.00)	1.00 (1.00–1.00)	**<0.001 ^2^**
Pain/Discomfort	1.00 (1.00–3.00)	1.00 (2.00–3.00)	1.00 (1.00–1.00)	**<0.001 ^2^**
Anxiety/Depression	1.00 (1.00–2.00)	1.00 (1.00–3.00)	1.00 (1.00–1.00)	**<0.001 ^2^**
EQ-5D-5L Index	0.78 (0.47–1.00	0.78 (0.52–1.00)	1.00 (0.91–1.00)	**<0.001 ^2^**
EQ-5D-5L EVA Score	60 (50–80)	70 (46.5–80)	80 (70–90)	**0.003 ^2^**

^1^ Fisher’s exact test; ^2^ Kruskal Wallis test1; EQ-5D-5L: Quality of Life Scale; IQR: Interquartile Range; Significant *p* values (<0.05) are in bold.

**Table 6 ijerph-19-02947-t006:** Multivariate analysis of the degree of acceptance of the use of new technologies.

		Middle Acceptability					High Acceptability	
	cOR (95% CI)	*p* Value	aOR ^a^ (95% CI)	*p* Value	cOR (95% CI)	*p* Value	aOR ^a^ (95% CI)	*p* Value
Sex								
Male	1.00 Ref.		1.00 Ref.		1.00 Ref.		1.00 Ref.	
Female	0.78 (0.44–1.41)	0.416	1.00 (0.51–1.96)	0.997	1.03 (0.59–1.80)	0.916	1.42 (0.68–2.96)	0.346
Age (1-year increase)	0.93 (0.90–0.97)	**<0.001**	0.94 (0.90–0.97)	**<0.001**	0.85 (0.82–0.89)	**<0.001**	0.88 (0.84–0.92)	**<0.001**
Education								
Uneducated	1.00 Ref.		1.00 Ref.		1.00 Ref.		1.00 Ref.	
Primary education	2.50 (0.94–6.64)	0.067	1.85 (0.62–5.48)	0.268	11.22 (3.98–31.71)	**<0.001**	7.23 (2.14–27.88)	**0.002**
Secondary education	1.20 (1.01–3.92)	**0.045**	2.82 (1.09–4.80)	**0.029**	6.22 (2.69–14.40)	**<0.001**	5.30 (1.90–14.78)	**0.001**
Care Expectations Fulfilled								
No	1.00 Ref.		1.00 Ref.		1.00 Ref.		1.00 Ref.	
Yes	0.57 (0.27–1.20)	0.139	0.66 (0.28–1.56)	0.340	0.64 (0.31–1.32)	0.231	0.65 (0.25–1.73)	0.392
Living Arrangements								
Residential center	1.00 Ref.		1.00 Ref.		1.00 Ref.		1.00 Ref.	
Live in company	1.15 (0.61–2.18)	0.659	0.62 (0.27–1.42)	0.262	14.24 (5.91–34.28)	**<0.001**	3.00 (1.00–9.08)	0.51
Alone	2.32 (0.97–5.52)	0.058	1.56 (0.54–4.47)	0.413	18.94 (5.92–34.29)	**<0.001**	5.11 (1.33–19.55)	**0.017**
Barthel (1 point increase)	0.99 (0.98–1.00)	0.267	0.99 (0.97–1.01)	0.300	1.05 (1.02–1.08)	**<0.001**	1.00 (0.98–1.02)	0.810
EQ-5D-5L (1 point increase)	0.71 (0.28–1.78)	0.463	0.76 (0.16–3.75)	0.740	21.70 (4.33–108.80)	**<0.001**	7.32 (0.71–76.01)	0.096
FES_I (1 point increase)	1.01 (0.97–1.06)	0.660	0.99 (0.94–1.06)	0.831	0.88 (0.83–0.94)	**<0.001**	0.94 (0.87–1.02)	0.148
VREM (1 point increase)	1.00 (1.00–1.00)	**0.025**	1.01 (1.00–1.02)	**0.039**	1.00 (1.00–1.00)	**<0.001**	1.00 (1.00–1.00)	0.236
SCQ (1 point increase)	1.08 (0.88–1.33)	0.471	1.10 (0.86–1.42)	0.439	0.77 (0.62–0.96)	**0.017**	1.17 (0.86–1.59)	0.327

cOR: crude odds ratio; aOR: adjusted odds ratio by multinomial logistic regression; CI: confidence interval; 95% IC: 95% confidence interval; ^a^: TAM in 3 categories as a dependent variable (low acceptance as reference category) and all other factors as independent variables in the model; EQ-5D-5L: quality of live index; FES_I: Fall risk scale (Downton) and fear of falling; SCQ: Self-Administered Comorbidity Questionnaire; VREM: Reduced version of the Minnesota Leisure Time Physical Activity Questionnaire; *p* values (<0.05) are in bold.

## Data Availability

Data from this study cannot be shared due to Data Custodian agreements relating to the access and use of the linked 4IE+ Study data held by the International Institute for Research and Innovation on Ageing. Code can be provided by the authors on request.

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
