# Peer review of "Factors Associated with the Acceptance of New Technologies for Ageing in Place by People over 64 Years of Age"

_ijerph, 2022, doi:10.3390/ijerph19052947_

Round 1
Reviewer 1 Report
This is an article based on sound research work. The only remark I have IS why the TAM model was chosen without a consideration of other models like the Unified Theory of Acceptance and Use of Technology (UTAUT) which gives a higher explanation of variance (Venkatesh et al. 2003).
- minor issues: - in the summary the suggestion is to leave the aOR values behind in the summary - introduction is broadly described, but all aspects are relevant. Suggestion to try to make it a little bit shorter in text not in the broadness - the use of more tables in the results section on page 6 and 7 would improve the overview for the reader - the motivation for the choice for the TAM model as the base for the research can be more convincing. Suggestion: put the TAM in the perspective of other models like UTAUT which gives a higher explanation for variance
Author Response
Responses to Reviewer 1:
This is an article based on sound research work. The only remark I have IS why the TAM model was chosen without a consideration of other models like the Unified Theory of Acceptance and Use of Technology (UTAUT) which gives a higher explanation of variance (Venkatesh et al. 2003).
We greatly appreciate the reviewer's comments.
Sentences have been added in lines 439-447 for clarification.
- In the summary the suggestion is to leave the aOR values behind in the summary
We have modified the sentences in lines 38-39.
- The use of more tables in the results section on page 6 and 7 would improve the overview for the reader
We fully agree with the reviewer. A new table 1 has been added and the text has been reduced to avoid redundancies.
- The motivation for the choice for the TAM model as the base for the research can be more convincing.
Thank you for this comment. We agree with the reviewer, and we have added a sentence pointing out this aspect in lines 439-447.
Reviewer 2 Report
General comment
This is a cross-sectional study to assess the degree of the acceptability of the use of new technologies and the factors associated with greater acceptance in older adults. In general, this is a relatively well-written article. I have minor comments as per below.
Background
- The relevant information has been provided.
Methods
- Page 4 Line 177 - please change the age of the population to be older than “64” instead of “65”.
- Page 4 Lines 200-208 - The TAM questions, which were used in this study to assess the degree of acceptability of the use of new technologies, are slightly different from the original one. If it was a modified one, please state it clearly. If the questionnaire and scoring system similar to the one used in this study has been used in the previous study, please cite the most similar one. And please translate the detail in the supplement file so the readers can read the questions.
- Page 5 – please include the information on how to interpret each questionnaire that has been used in this study, for example, FES-I, VREM, Barthel ADL index, and EQ-5D.
- Page 6 line 279 – please change “ORs” to “aORs”
- Data analysis – please add the p-value that was used to consider statistically significant. Was it 0.05?
Results
- Page 6 – 9 - Please make sure that the authors did not read the tables. The statements can be shorter by stating only important information from tables 1-3.
Discussion
- From the result, about 35 percent has a low readiness for the use of new technologies. It would be nice to also see the interventions that would help increase readiness in this population.
Author Response
This is a cross-sectional study to assess the degree of the acceptability of the use of new technologies and the factors associated with greater acceptance in older adults. In general, this is a relatively well-written article. I have minor comments as per below.
Methods
Page 4 Line 177 - please change the age of the population to be older than “64” instead of “65”
We thank the reviewer. The error has been corrected
Page 4 Lines 200-208 - The TAM questions, which were used in this study to assess the degree of acceptability of the use of new technologies, are slightly different from the original one. If it was a modified one, please state it clearly. If the questionnaire and scoring system similar to the one used in this study has been used in the previous study, please cite the most similar one. And please translate the detail in the supplement file so the readers can read the questions.
The paragraph has been modified to clarify this aspect and the English translation of the questionnaire used has been added so that it can be consulted in the supplement file.
Page 5 – please include the information on how to interpret each questionnaire that has been used in this study, for example, FES-I, VREM, Barthel ADL index, and EQ-5D.
Information on interpretation has been added to some of the scales that lacked it. (Lines 228-229/ 238-240/ 246-247)
Page 6 line 279 – please change “ORs” to “aORs”
We apologize for this mistake. It has been corrected.
Data analysis – please add the p-value that was used to consider statistically significant. Was it 0.05?
We thank the reviewer. We have added it in line 287
Results
Page 6 – 9 - Please make sure that the authors did not read the tables. The statements can be shorter by stating only important information from tables 1-3.
We thank the reviewer for his comment. We have taken care that only the information we want to highlight from the tables has been included in the text.
Discussion
From the result, about 35 percent has a low readiness for the use of new technologies. It would be nice to also see the interventions that would help increase readiness in this population.
We fully agree with the reviewer. We have reflected on this in lines 473-480 of the discussion.
Reviewer 3 Report
Some comments are suggested:
- In the abstract, the design used must be indicated. If it has been a descriptive or analytical observational study.
- It would be interesting if the keywords are DeCS/MeSH descriptors.
- The introduction is complete and well-founded to explain and justify the phenomenon under study
- It has been indicated that a pilot study has been carried out with 80 participants. Why that number? It is somewhat high for a pilot test.
- Abbreviations must be clarified in the text the first time they appear (for example TAM, PU, ATU, etc.)
- The instruments must provide the validation data obtained by all scales.
- The study has not been authorized by a Research Ethics Committee? It would not be sufficient for the review committee of the university unless it has the character of an approved research ethics committee
- It is recommended in the first three paragraphs of the results, when the sociodemographic characteristics are explained, that these be subdivided more clearly to improve the readability of said data.
- On some occasions, the same information is repeated in the text and in the tables. Check to avoid it and reduce the quality of the results.
- The discussion and conclusions are consistent with the results obtained.
Author Response
Some comments are suggested:
In the abstract, the design used must be indicated. If it has been a descriptive or analytical observational study. It would be interesting if the keywords are DeCS/MeSH descriptors.
We greatly appreciate the reviewer's comments. This information has been added to the abstract.
The introduction is complete and well-founded to explain and justify the phenomenon under study
It has been indicated that a pilot study has been carried out with 80 participants. Why that number? It is somewhat high for a pilot test.
We wanted to ensure that we had sufficient quality data due to the large amount of information collected by each questionnaire for this study and the high attrition of this population group.
Abbreviations must be clarified in the text the first time they appear (for example TAM, PU, ATU, etc.)
Abbreviations throughout the manuscript have been revised.
The instruments must provide the validation data obtained by all scales.
We are grateful for your comments. This information is available on the following lines 211-268
The study has not been authorized by a Research Ethics Committee? It would not be sufficient for the review committee of the university unless it has the character of an approved research ethics committee
The project was approved by the Research Ethics Committee of the University of Extremadura. The translation was mistakenly changed to "review committee". This has been corrected on line 291. We apologise for the mistranslation.
It is recommended in the first three paragraphs of the results, when the sociodemographic characteristics are explained, that these be subdivided more clearly to improve the readability of said data.On some occasions, the same information is repeated in the text and in the tables. Check to avoid it and reduce the quality of the results.
We fully agree with your comment, which is in line with those of the other reviewers. This is why we have restructured the beginning of the introduction by introducing a new table and shortening the text to make it easier to read.
The discussion and conclusions are consistent with the results obtained.
We are grateful for your comments, and we hope that you consider the changes and improvements made to the manuscript to be appropriate